# LiteMedSAM with Low-Rank Adaptation and Multi-Box Efficient Inference for Medical Image Segmentation

Wentao Liu[1][0000−0003−0837−5555], Weijin Xu[1][0000−0001−8371−8330], Ruifeng Bian[1][0009−0003−7043−4023], Haoyuan Li[1][0009−0002−1176−3583], and Tong Tian[2][0009−0009−0039−244X]

[1] School of Artificial Intelligence, Beijing University of Posts and Telecommunications, Beijing 100876, China
[2] State Key Laboratory of Structural Analysis, Optimization and CAE Software for Industrial Equipment, School of Aeronautics and Astronautics, Dalian University of Technology, Dalian 116024, China
[3] School of Computer Science and Information Security, Guilin University of Electronic Technology, Guilin, China
liuwentao@bupt.edu.cn

**Abstract.** Medical image segmentation is essential in clinical practice for accurately quantifying anatomical structures and pathological regions. Despite a shift towards foundation models capable of handling various segmentation tasks, current models are often optimized for natural images and require substantial computational resources, limiting their widespread clinical use. In this paper, we analyze the distribution of data across different modalities of medical images in the training dataset of the challenge. We adjust the probabilities of selecting each modality during data loading to alleviate the severe imbalance in modality data and improve the segmentation performance of medical images with limited data in certain modalities. We fine-tune LiteMedSAM, incorporating the low-rank adaptation technique into the multi-head attention and multilayer perceptron of TinyVit. To improve inference speed, we concurrently perform inference with multiple box prompts and utilize the argmax operation to process the outputs of multiple box prompts, thereby enhancing segmentation accuracy.

**Keywords:** Medical image segmentation · Segment Anything Model · Low-Rank Adaptation.

## 1 Introduction

Medical image segmentation serves as a crucial component in clinical practice, enabling the precise quantification of anatomical structures and pathological regions. Currently, the field is undergoing a significant transformation, shifting from specialized models tailored to specific tasks towards more versatile foundation models capable of handling diverse segmentation scenarios. One notable

example is the Segment Anything Model (SAM) [1], a family of image segmentation models pretrained on a comprehensive dataset comprising 11 million images and 1 billion masks. SAM exhibits remarkable zero-shot image segmentation performance and finds applications across various domains, including medical image segmentation.

Despite these advancements, a prevalent challenge persists: many existing segmentation foundation models are optimized primarily for natural images or demand substantial computational resources during inference. This limitation poses a significant obstacle to their widespread adoption within clinical settings. Therefore, there is a pressing need to refine and adapt these models to meet the unique demands of medical imaging, ensuring their practical feasibility and effectiveness in clinical applications.

This challenge aims to develop universal, promptable medical image segmentation models that can be deployed on laptops or other edge devices without relying on GPUs. Specifically, the task involves creating a lightweight bounding box-based segmentation model. To support this endeavor, This challenge offers a comprehensive training dataset containing over 1,000,000 image-mask pairs. This dataset covers 10 medical image modalities and encompasses more than 20 types of cancer. The goal is to develop models that not only deliver accurate segmentation results but also operate efficiently on resource-constrained devices, facilitating their widespread deployment in clinical settings.

To accelerate SAM, numerous efforts have been made to replace SAM's image encoder with lightweight models. For example, MobileSAM [7] distills the knowledge of SAM's ViT-H model into a tiny vision transformer. EdgeSAM [**?**] trains a purely CNN-based model to mimic ViT-H, employing a meticulous distillation strategy with the prompt encoder and mask decoder involved in the process. EfficientSAM [5] leverages the MAE pretraining method to improve performance. EfficientViT-SAM mitigates performance decline by replacing SAM's image encoder with EfficientViT [8] while reducing computation costs. In the domain of medical image segmentation, MedSAM, a refined foundational model, significantly enhances SAM's segmentation performance on medical images. This accomplishment is achieved through fine-tuning SAM on an unprecedented dataset containing over one million medical image-mask pairs. Further, a lightweight version of MedSAM (LiteMedSAM) is proposed by replacing MedSAM's image encoder with TinyVit to improve inference speed for deployment on laptops.

In this paper, we analyze the distribution of data across different modalities of medical images in the training dataset. We adjust the probabilities of selecting each modality during data loading to alleviate the severe imbalance in modality data and improve the segmentation performance of medical images with limited data in certain modalities. We fine-tune LiteMedSAM, incorporating the low-rank adaptation technique into the multi-head attention and multilayer perceptron of TinyVit. To improve inference speed, we concurrently perform inference with multiple box prompts and utilize the argmax operation to process the outputs of multiple box prompts, thereby enhancing segmentation accuracy.

## 2    Method

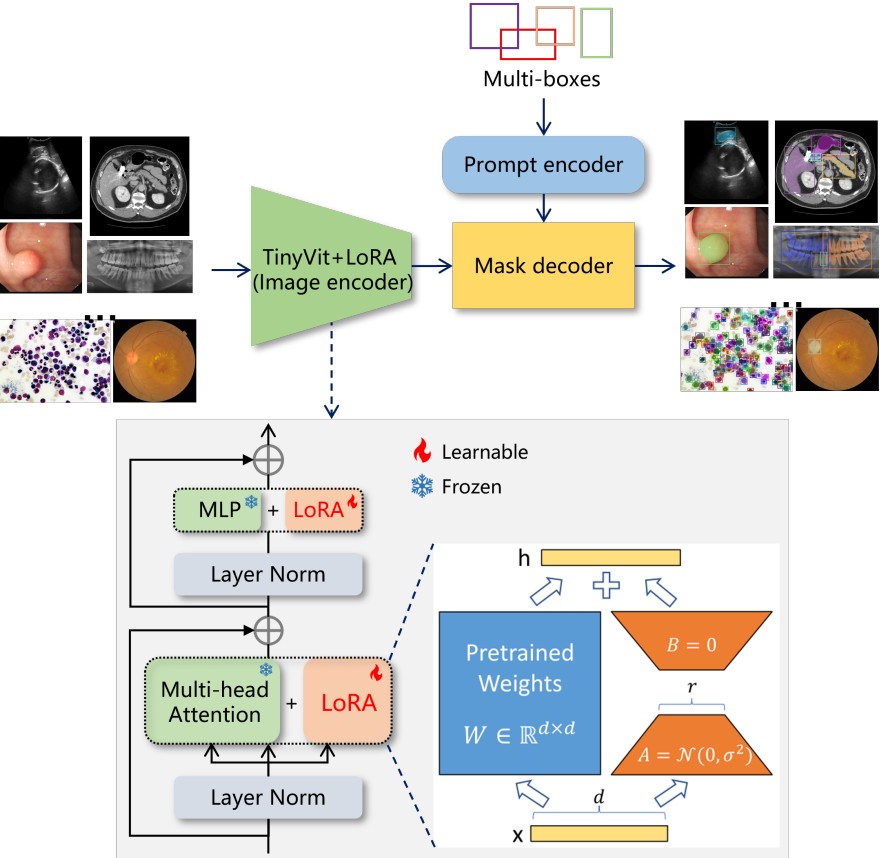

**Fig. 1.** Network architecture (Copyright preserved. Please do not directly use this figure in your manuscript.) Please also include the network description in the figure title. So reviewers could quickly understand your idea.

### 2.1    Preprocessing

Based on the distribution of data quantities, the various medical image modalities show significant disparities. CT (Computed Tomography) has the largest dataset, with 1,218,411 samples, comprising nearly 82% of the total. The next largest is MR (Magnetic Resonance), with 191,308 samples, accounting for approximately 13%. Other medical image modalities have relatively smaller datasets; for instance, Microscopy, Fundus, Mammography, US, and OCT each have fewer

than 2,000 samples. Their respective proportions are all below 3%, with many even less than 1%. This distribution could significantly impact model training, potentially resulting in poorer segmentation performance for medical image modalities with smaller datasets.

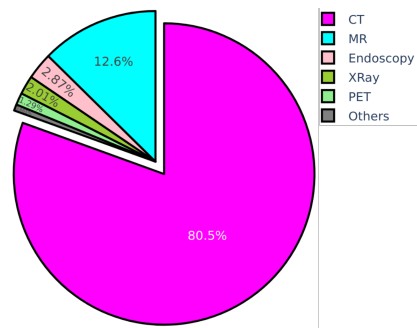

**Fig. 2.** Distribution of various medical image modality quantities in the training set

To address this issue, during training, we randomly select data from different medical image modalities based on their quantities. Specifically, CT, having the largest quantity, will be selected with a probability of 40%. MR, being the second largest, will be selected with a probability of 10%. The remaining 50% of the time, we will randomly choose from the other nine modalities, which have smaller quantities. The data processing involves resampling based on the largest dimension, followed by min-max normalization and padding with zeros to make the height and width consistent. Data augmentation includes random horizontal and vertical flipping.

## 2.2 Proposed Method

**Please provide figures to show your pipeline or network architecture.**
Figure 2 shows a typical example of 3D U-Net

In order to reduce SAM's parameters and improve inference speed, the image encoder of SAM adopts the lightweight VIT variant, TinyVIT [4]. TinyVIT adopts a hierarchical vision transformer as its fundamental architecture, for the convenience of dense prediction downstream tasks like segmentation that require multi-scale features. Specifically, the base model comprises four stages, with a gradual reduction in resolution akin to previous works such as Swin and LeViT . The patch embedding block consists of two convolutions with a kernel size of 3, a stride of 2, and padding of 1. Lightweight and efficient MBConvs are employed in Stage 1 and downsampling blocks, leveraging the effectiveness of convolutions at earlier layers in efficiently learning low-level representations due to their strong inductive biases. The last three stages are constructed using transformer blocks, incorporating window attention to reduce computational cost. Attention biases

and a $3 \times 3$ depthwise convolution between attention and MLP are introduced to capture local information. Residual connections are applied to each block in Stage 1, as well as attention blocks and MLP blocks. Activation functions throughout the model employ GELU. Convolutional and linear normalization layers utilize BatchNorm and LayerNorm , respectively.

Low-Rank Adaptation (LoRA) is a technique designed to enhance the efficiency of neural networks by reducing the number of trainable parameters. When applied to the Segment Anything Model (SAM) for medical image segmentation, LoRA decomposes the weight matrices within SAM's architecture into lower-rank matrices. This reduction in computational and memory requirements allows SAM to be fine-tuned effectively on large medical image datasets, such as those used for segmenting anatomical structures and pathological regions. By incorporating LoRA, SAM can maintain high segmentation performance while being more resource-efficient, facilitating its deployment in clinical settings where computational resources are often limited. We fine-tune LiteMedSAM, incorporating the low-rank adaptation technique into the multi-head attention and multilayer perceptron of TinyVit.

Loss function: we use the summation between Dice loss and focal loss because compound loss functions have been proven to be robust in various medical image segmentation tasks [2].

To improve inference speed, we concurrently perform inference with multiple box prompts and utilize the argmax operation to process the outputs of multiple box prompts, thereby enhancing segmentation accuracy.

### 2.3 Post-processing

We did not use any post-processing techniques.

## 3 Experiments

### 3.1 Dataset and evaluation measures

We only used the data provided by the challenge for training.

The evaluation metrics include two accuracy measures—Dice Similarity Coefficient (DSC) and Normalized Surface Dice (NSD)—alongside one efficiency measure—running time. These metrics collectively contribute to the ranking computation.

### 3.2 Implementation details

**Environment settings** The development environments and requirements are presented in Table 1.

**Table 1.** Development environments and requirements. (mandatory table)

| | |
|---|---|
| System | Ubuntu 18.04.5 LTS |
| CPU | Intel(R) Core(TM) i9-7900X CPU@3.30GHz |
| RAM | 16×4GB; 2.67MT/s |
| GPU (number and type) | One NVIDIA A100 80G |
| CUDA version | 11.0 |
| Programming language | Python 3.20 |
| Deep learning framework | torch 2.0, torchvision 0.2.2 |
| Specific dependencies | |
| Code | https://github.com/lseventeen/SAMIL |

**Training protocols** During training, we randomly select data from different medical image modalities based on their quantities. Specifically, CT, having the largest quantity, will be selected with a probability of 40%. MR, being the second largest, will be selected with a probability of 10%. The remaining 50% of the time, we will randomly choose from the other nine modalities, which have smaller quantities. The data processing involves resampling based on the largest dimension, followed by min-max normalization and padding with zeros to make the height and width consistent. Data augmentation includes random horizontal and vertical flipping.

**Table 2.** Training protocols. (mandatory table)

| | |
|---|---|
| Pre-trained Model | LiteMedSAM [3] |
| Batch size | 16 |
| Patch size | 256×256 |
| Total epochs | 20 |
| Optimizer | AdamW |
| Initial learning rate (lr) | 0.00005 |
| Lr decay schedule | ReduceLROnPlateau |
| Training time | 30 hours |
| Loss function | Dice loss and focal loss |
| Number of model parameters | 985.22M[4] |
| Number of flops | 59.32G[5] |
| $CO_2$eq | 1 Kg[6] |

## 4   Results and discussion

Note: Please describe at least the following aspects in this section
    - In what kind of cases the proposed method works well?
    - What are the possible reasons for the failed cases?

- Segmentation efficiency analysis

**Table 3.** Quantitative evaluation results. **The last two columns should correspond to your final docker submission**. Please show at least one ablation study result A useful online tool to create latex table [https://www.tablesgenerator.com/latex_tables.](https://www.tablesgenerator.com/latex_tables) (mandatory table)

| Target | Baseline | | Ablation Study 1 | | Ablation Study 2 | | Proposed | |
|---|---|---|---|---|---|---|---|---|
| | DSC(%) | NSD(%) | DSC(%) | NSD(%) | DSC(%) | NSD (%) | DSC(%) | NSD (%) |
| CT | 92.26 | 94.9 | 89.67 | 91.66 | 92.47 | 95.3 | 92.03 | 94.18 |
| MR | 89.63 | 93.37 | 82.12 | 84.67 | 87.92 | 91.89 | 86.39 | 89.5 |
| PET | 51.58 | 25.17 | 69.86 | 52.35 | 62.64 | 42.03 | 65.23 | 44.91 |
| US | 94.77 | 96.81 | 83.7 | 88.52 | 85.61 | 90.45 | 87.17 | 92.35 |
| X-Ray | 75.83 | 80.39 | 76.63 | 82.29 | 85.08 | 90.1 | 84.26 | 89.23 |
| Dermotology | 92.47 | 93.85 | 94.92 | 96.34 | 94.69 | 96.13 | 94.8 | 96.17 |
| Endoscopy | 96.04 | 98.11 | 96 | 98.25 | 96.08 | 98.44 | 96.57 | 98.75 |
| Fundus | 94.81 | 96.41 | 96.01 | 97.53 | 95.95 | 97.53 | 95.96 | 97.51 |
| Microscopy | 61.63 | 65.38 | 81.35 | 87.78 | 79.64 | 85.88 | 81.07 | 87.33 |
| Average | 83.23 | 82.71 | 85.58 | 86.6 | 86.68 | 87.53 | 87.05 | 87.77 |

Note to Table 3: if you have multiple solutions, such as a faster model with lower DSC or a slower model with higher DSC, you can use a similar Table format to report the performance on the public/online validation set.

### 4.1   Quantitative results on validation set

Please describe the results

### 4.2   Qualitative results on validation set

please show some examples with good segmentation results and two examples with bad segmentation results.

Note: the ground truth of the validation set is not available but authors can show results on other public datasets where annotations are available

### 4.3   Segmentation efficiency results on validation set

### 4.4   Results on final testing set

This is a placeholder. We will announce the testing results during CVPR (6.17-18)

8      W. Liu et al.

**Table 4.** Quantitative evaluation of segmentation efficiency in terms of running time (s). Note: The inference process cannot use GPU. If you didn't make validation docker submissions during the challenge, you can obtain these metrics on your local laptop. (mandatory table)

| Case ID | Size | Num. Objects | Baseline | Ablation Study | Proposed |
|---|---|---|---|---|---|
| 3DBox_CT_0566 | (287, 512, 512) | 6 | 376.4 | | |
| 3DBox_CT_0888 | (237, 512, 512) | 6 | 100.5 | | |
| 3DBox_CT_0860 | (246, 512, 512) | 1 | 17.7 | | |
| 3DBox_MR_0621 | (115, 400, 400) | 6 | 157.1 | | |
| 3DBox_MR_0121 | (64, 290, 320) | 6 | 99.9 | | |
| 3DBox_MR_0179 | (84, 512, 512) | 1 | 17.1 | | |
| 3DBox_PET_0001 | (264, 200, 200) | 1 | 12.1 | | |
| 2DBox_US_0525 | (256, 256, 3) | 1 | 6.3 | | |
| 2DBox_X-Ray_0053 | (320, 640, 3) | 34 | 7.3 | | |
| 2DBox_Dermoscopy_0003 | (3024, 4032, 3) | 1 | 6.5 | | |
| 2DBox_Endoscopy_0086 | (480, 560, 3) | 1 | 6.1 | | |
| 2DBox_Fundus_0003 | (2048, 2048, 3) | 1 | 6.1 | | |
| 2DBox_Microscope_0008 | (1536, 2040, 3) | 19 | 6.8 | | |
| 2DBox_Microscope_0016 | (1920, 2560, 3) | 241 | 19.1 | | |

### 4.5 Limitation and future work

## 5 Conclusion

In this paper, we analyze the distribution of data across different modalities of medical images in the training dataset. We adjust the probabilities of selecting each modality during data loading to alleviate the severe imbalance in modality data and improve the segmentation performance of medical images with limited data in certain modalities. We fine-tune LiteMedSAM, incorporating the low-rank adaptation technique into the multi-head attention and multilayer perceptron of TinyVit. To improve inference speed, we concurrently perform inference with multiple box prompts and utilize the argmax operation to process the outputs of multiple box prompts, thereby enhancing segmentation accuracy.

**Acknowledgements** We thank all the data owners for making the medical images publicly available and CodaLab [6] for hosting the challenge platform.

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

**Table 5.** Checklist Table. Please fill out this checklist table in the answer column.

| Requirements | Answer |
| --- | --- |
| A meaningful title | Yes/No |
| The number of authors ($\leq$6) | Number |
| Author affiliations and ORCID | Yes/No |
| Corresponding author email is presented | Yes/No |
| Validation scores are presented in the abstract | Yes/No |
| Introduction includes at least three parts: background, related work, and motivation | Yes/No |
| A pipeline/network figure is provided | Figure number |
| Pre-processing | Page number |
| Strategies to data augmentation | Page number |
| Strategies to improve model inference | Page number |
| Post-processing | Page number |
| Environment setting table is provided | Table number |
| Training protocol table is provided | Table number |
| Ablation study | Page number |
| Efficiency evaluation results are provided | Table number |
| Visualized segmentation example is provided | Figure number |
| Limitation and future work are presented | Yes/No |
| Reference format is consistent. | Yes/No |
| Main text >= 8 pages (not include references and appendix) | Yes/No |