# OpenReview forum: "LiteMedSAM with Low-Rank Adaptation and Multi-Box Efficient Inference for Medical Image Segmentation"
_thecvf.com/CVPR/2024/Workshop/MedSAMonLaptop — Submitted to CVPR24 MedSAMonLaptop_

### Official Review · Reviewer_MFoz · 2024-06-11
**Review: The Paper is incomplete.**

**Rating:** 4
**Confidence:** 5

**Review:**

The authors' ideas in the abstract, introduction, method, and conclusion sound very interesting. Moreover, they explain the idea clearly.

However, the authors did not update the result and discussion section (Section 4) from the template. Essentially, Section 4 is missing any new content or the validation results. Moreover, Table 5, which can be understood as the submission checklist, is also not filled out. As far as I understand, both should have been filled out by now, and it would be important to understand the impact of the contribution.

### Other Feedback:

* In the introduction, the reference for EdgeSAM and the reference (GitHub Link) for LiteMedSAM are missing.
* The caption in Figure 1 needs to be updated. Otherwise, the overview diagram is very nice!
* In Section 2.1, a table showing all the numbers described in the text would be very beneficial.
* Regarding the method described in Section 2.1, the relation to stratified sampling/splits and/or the application thereof might be beneficial to put your method into context.
* In Section 2.3, the first sentences are still from the template.
* In Section 2.3., the reference for LoRA is missing.
* Providing more details on how you perform inference concurrently in the CPU setup might be beneficial.
* In Table 1, Python 3.20 is likely a typo.
* The provided code could benefit from a better readme that reflects the methods described in this paper.
* The footnotes in Table 2 are broken. I think this is a problem with the table environment.

---

### Official Review · Reviewer_KgBr · 2024-06-13
**The paper is marginally below acceptance treshold, since a big part of the result is missing and there are a couple of other inaccuracies.**

**Rating:** 5
**Confidence:** 3

**Review:**

The main claim is that using a finetuned version of LiteMedSAM, that was created by sampling different modalities with specific probabilities and using LoRA, is able to outperform the baseline when running inference with multiple boxes in parallel. The authors used low-rank adaption for the TinyViT image encoder of LiteMedSAM.
They decided to sample CT datapoints with 40% probability, MR with 10% probability and all other 9 modalities with 50% probability. The authors achieved an average dice similarity coefficient of 87.05% and normalized surface distance of 87.77% beating the baseline on the validation set.

The paper seems to be marginally below the acceptance threshold, it proposes a fine tuning pipeline based on LoRA and sampling modalities with different probabilities, it presents an inference system based on inferencing with multiple boxes in parallel, but it misses a lot of information in the results section, such as the efficiency results on the validation set, which makes it hard to judge how well the proposed approach performs.

## Clarity
It was not clear to me what you meant with multi-processing the boxes and using argmax, I had to read the code, I would recommend clarifying what you did by mentioning that you did batched inference for the prompt encoder and mask decoder for 2D datapoints with multiple boxes.
You mention that 50% of the time you will randomly choose from the other 9 modalities. It would be good to clarify whether the 9 other modalities will be sampled with equal probability or proportional to their number of datapoints.
LoRA makes finetuning more ressource efficient, but in Section 2.2 you mention that LoRA makes deployment more ressource efficient?
It is also not clear to me why concurrently performing inference with multiple boxes is supposed to enhance the segmentation accuracy, as mentioned in the Introduction and abstract. Maybe you meant segmentation efficiency?
Table 3 contains two ablation studies, but there is no information what they were about.

## Reproducibility
The paper contains some information about the environment used, but a lot seems to be copied over from the template, including the yet to be released Python 3.20,
it would be good to add more accurate information about the environment used.
The GitHub repository also lacks a README with instructions on how to reproduce the results as well as environment information like a requirements.txt.

## Typo
In table 2 the number of parameters seem to be off by an order of magnitude. And the number of flops and CO2eq are copied from the template, I would recommend leaving these values out if you did not compute them.

---

### Official Review · Reviewer_JBKM · 2024-06-15
**Good network but missing so many parts**

**Rating:** 3
**Confidence:** 3

**Review:**

the paper has a good abstract, introduction and a clear graphy for their method of ViT+LoRA. But there are many parts missing in the following parts including  postprocessing,Quantitative results on validation set,Qualitative results on validation set.

---

### Official Review · Reviewer_H5i8 · 2024-06-16

**Rating:** 3
**Confidence:** 5

**Review:**

### Paper Summary
This paper implements LORA and fin-tuning to achieve inference of medical images segmentation on laptop. The result is better than the baseline LiteMedSAM.
### Paper Strength
The introduction and method are relatively complete. The result is promising.
### Paper Weakness
The paper is not complete in discussion.  There is no qualitative result, no discussion, no conclusion. Simply speaking, the paper was not finished.

---

### Decision · Program_Chairs · 2024-10-01

**Decision:**

Major Revision

**Comment:**

Please address the concerns of all reviewers and add testing results. Otherwise, the paper will be rejected in the last round.